# Reasoning action-centric temporal relations at rich feature hierarchies for action recognition

**Manshu Liang** [iD][1*], **Wenbin Wu**[1], **Zhuolei Chen**[1], **Tengfei Han**[1], **Yuan Zheng**[2]

**1** Electric Power Science Research Institute, State Grid Fujian Electric Power Co. Ltd., Fujian, China,
**2** School of Computer Science, Civil Aviation Flight University of China, Deyang, China

* amanda498124338@gmail.com

**Data availability statement:** Data are located at the following: Something-Something datasets

## Abstract

Reasoning temporal relations among action-related objects plays an important role in action recognition. However, previous approaches only focus the reasoning on high-level semantics and inevitably involve the background in reasoning. In this work, we propose to formulate the temporal relational reasoning in an action-centric and hierarchical style, with a novel Action-centric Temporal-relational Reasoning (ATR) block. Specifically, ATR comprises two components: an Action-related Region Locator (ARL) to locate the action-related regions via estimating the actionness, and an Efficient Action-centric Reasoner (EAR) to efficiently reason the temporal relations between the located regions so as to realize the action-centric reasoning. Thanks to its flexible and efficient designs, our ATR can be directly integrated into existing action recognition models at different depths, empowering the hierarchical reasoning on the action-centric temporal relations at the cost of minor computational overhead. We extensively evaluate our ATR block on three action recognition benchmarks, Something-Something V1, V2, and Kinetics, with the backbones of TSN, TSM, and SlowOnly. The consistent and notable improvements over the strong baselines sufficiently corroborate the effectiveness of ATR, along with the action-centric and hierarchical formulation for temporal relational reasoning. Our proposed approach provides potential practical significance for real-world scenarios.

## Introduction

Viewing that our world is live, it is important to extend image understanding [1–4] to video understanding [5–8], which provides rich information for downstream applications. As a fundamental task in video understanding, action recognition [9–12]recently has attracted increasing interest in academic and industrial communities [13,14]. Albeit the substantial advance, this task is still very challenging. One critical difficulty lies in temporal relational reasoning, which is the ability to link action-related objects over time and understand their transformation process.

download link: https:
//www.qualcomm.com/developer/software/
something-something-v-2-dataset/downloads.
Kinetics datasets download link: https:
//github.com/cvdfoundation/kinetics-dataset.

**Funding:** The author(s) received no specific
funding for this work.

**Competing interests:** The authors have
declared that no competing interests exist.

In literature, convolutional neural networks (CNNs) [5,15,16] are widely used to address action recognition. However, as shown in [17], the ability of reasoning action's temporal relations is limited when directly applying CNNs. To tackle this problem, a Temporal Relation Network (TRN) is introduced [17] as the reasoning step in Fig 1(a), to reason the temporal relations between high-level features. Despite being interpretable, TRN is still subject to two deficiencies. *First*, it lacks reasoning on low-level features. As demonstrated in [18–20], low-level features can encode object details, hence are beneficial to capture the fine-grained object transformation over time. *Second*, the reasoning inevitably suffers from the background distraction problem, since its input is the average of all information including the background. The complicated background, as observed in [21–23], can lead to a strong action recognition bias.

A natural approach to alleviating the background distraction problem in action recognition is to locate action-related regions and focus temporal reasoning on them. This idea has been explored in prior works [24–27] where object detectors are employed as region locators, as illustrated in Fig 1(b). However, these detectors often produce all visible objects, including those irrelevant to the action, thereby failing to fully eliminate background interference during reasoning. Moreover, existing methods such as TRN primarily rely on high-level semantic features while neglecting the potential benefits of low-level representations. Their computationally intensive and rigid structures also limit their applicability to multi-level feature hierarchies.

Motivated by these limitations, we propose to formulate temporal relational reasoning in an action-centric and hierarchical manner. Our goal is to (1) accurately localize truly action-relevant regions to suppress background noise, and (2) enable efficient multi-level integration to fully exploit both low- and high-level spatiotemporal features. To this end, we propose a novel built-in style block dubbed Action-centric Temporal-relational Reasoning (ATR). As shown in Fig 1(c), our ATR is basically constitutive of two components, i.e., an Action-related Region Locator (ARL) and an Efficient Action-centric Reasoner (EAR). Specifically, ARL locates action-related regions via estimating the actionness of each spatial position, which is accomplished by jointly modeling motion and appearance at the feature-level. With the actionness maps obtained from ARL, EAR first action-centrically aggregates feature maps to be temporal-wise feature vectors. Temporal relations are next reasoned on them by a new

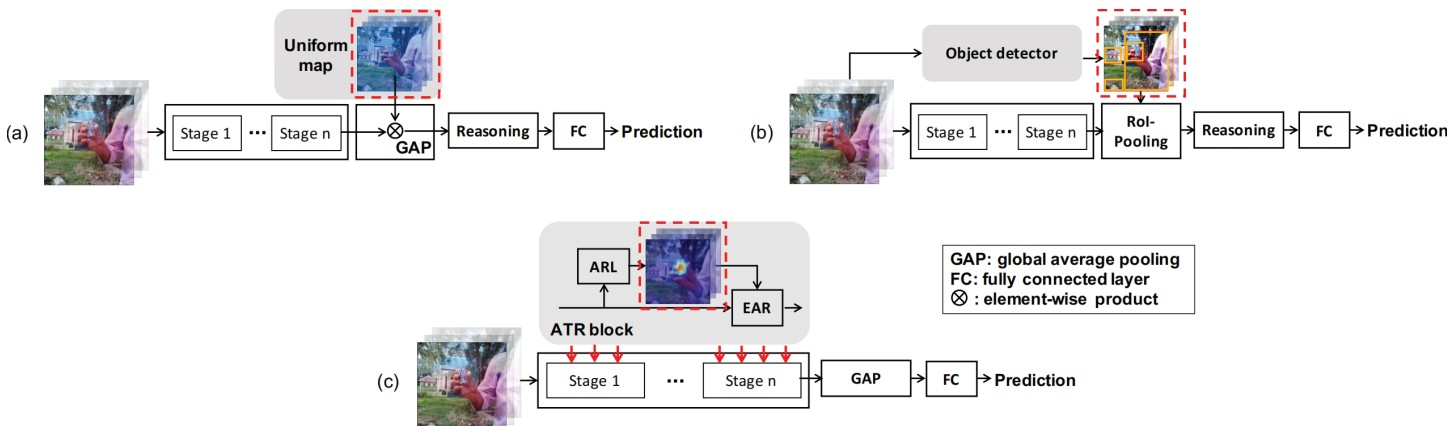

**Fig 1. Previous temporal relations reasoning pipelines, (a) and (b), suffer from the problems of background distraction and lack of reasoning on multi-level semantics.** Our proposed (c) ATR-nets solve them by reasoning temporal relations in an action-centric and hierarchical formulation.

efficient reasoner, which is simplified from the previous graph-based reasoner [24] under the guidance of empirical observations. The reasoned results are finally distributed back to the action-related regions. With this action-centric feature aggregation and distribution to integrate the ARL and the reasoner, the temporal relations can thus be action-centrically reasoned by our ATR.

By virtue of the overall efficient and built-in style designs, our proposed ATR block can be integrated into various architectures, in a plug-and-play manner with only minor overhead introduced. When being equipped at multiple depths, the transformations of action-related objects are comprehensively perceived from rich-level semantics. Extensive experiments on Kinetics [28] and Something-Something V1 & V2 [29] datasets verify that our hierarchical, action-centric formulation can significantly improve the performance. In particular, various action recognition models, such as TSN [5], TSM [30] and SlowOnly [31], attain remarkable performance boosts after integrating ATR. When compared to other state-of-the-art temporal relational reasoning methods, we also demonstrate explicit superiority in both accuracy and efficiency aspects.

To summary, our contributions are four folds: (1) We develop an actionness estimator in play-and-plug style, to locate the action-related regions. (2) We propose a lightweight reasoner to reason the temporal relations. (3) By integrating the actionness estimator and the reasoner with the action-centric feature aggregation and distribution, our ATR realizes the action-centric and hierarchical formulation for temporal relational reasoning, which is the first to our best known. (4) Extensive experiments validate that our action-centric and hierarchical formulation is critical to reason action's temporal relations. Besides, various SOTA 2D/3D models achieve notable improvements after equipping ATR, providing a worthy insight to the community: The ability of reasoning temporal relations should be better considered in future model design.

## Related work

**Video action recognition.** With the rapid development of deep convolutional networks, CNN-based methods are dominating in action recognition. One of the most popular paradigms is in a two-stream architecture, where two 2D-CNNs are utilized to separately model appearance and motion information [32–34]. Built upon it, many extensions have been raised to further improve the performance. For instance, TSN [5] introduces a sparse and uniform sampling strategy to better collect a video clip. On the other hand, the CNN-LSTM framework has been widely applied for video action recognition [35,36]. In this framework, CNN is used to extract spatial features for each frame, and then LSTM is employed to represent temporal features from feature maps. For instance, ResLNet [37] proposes a novel deep residual LSTM network, which can take longer inputs and have convolutions collaborate with LSTM more effectively under the residual structure to learn better spatial-temporal representations.

Lately, 3D-CNN-based methods have become preferable for their superior performances [38–40]. A key point lies in expanding the 2D convolution with temporal dimension, hence spatial and temporal semantics can be jointly learned. However, the simple expansion brings heavy overheads, leading the network to be hard to train. To address this issue, plenty of methods have been proposed, through inflating 2D convolutional kernels [15], spatio-temporal factorizing [16,41–43], etc. TSM [30], further bridges the efficiency and performance gaps between 2D-CNN based and 3D-CNN-based methods, by directly combining the features from consecutive frames. Recently, STAN [44] introduces improved 2D-CNNs with Temporal Embedding Head and Spatio-Temporal Attention components, allowing the model

to capture subtle temporal changes and achieve performance comparable to that of 3D-CNNs and Transformers.

**Temporal relational reasoning.** The concept of temporal relational reasoning is first introduced in [17], which develops the TRN module to conquer the deficiency of CNN-based methods in reasoning temporal relations. However, two fundamental problems exist in TRN, which are background distraction and lack of multi-level reasoning. An alternative pipeline is via *detecting + graph-based reasoning*: Objects are first located by an object detector, and their relations are then reasoned by a graph-based reasoner [24]. However, many action-unrelated objects are inevitably involved, distracting the reasoning process. To alleviate this problem, STIN [25] separates the detection of the subject (or agent) and objects in an action, and restricts the object number to be reasoned; MUSLE [27] class-specifically explores the discriminative sub-graphs out of the whole graph. Nevertheless, tremendous overhead is introduced for detecting and reasoning. And the same as TRN, only high-level features are explored and reasoned. Nie *et al* propose an adaptive solid-state synapse with bi-directional relaxation for spatio-temporal pattern recognition [45].

Different from the above works, we reason temporal relations in the action-centric and hierarchical formulation. More specifically: 1) To locate action-related regions, we explore where the action occurs (actionness) but not where objects exist (objectness), so that the located regions are more related to the action itself. 2) To reason temporal relations, we advance a novel lightweight reasoner, which is more efficient and performs better than previous ones.

**Actionness estimation.** Chen *et al.* first raise the notion of actionness [46]. Since then, actionness estimation is broadly studied because of its benefits for video-based tasks. To better estimate the actionness, various methods have been proposed. Luo *et al.* introduce meaningful attributes for actionness estimation, which reveal the agency and intentionality of an action [47]. Li *et al.* [48] and Yu *et al.* [49] propose to calculate actionness based on the human detection score and motion score. The most related work to ours is H-FCN [50], where two FCNs are adopted respectively with RGB and optical flow as inputs, to estimate actionness maps from appearance and motion aspects. TASTA [51] introduces a novel spatial and temporal attention network that estimates the attention spatially and temporally for capturing video feature extraction with textual information as an assistant. Our work differs from theirs in that we estimate actionness at the feature level. Moreover, only lightweight operations are used, enabling our actionness estimator to be unified into the same framework with spatio-temporal feature learning.

## Method

### Overview

Our goal is to perform temporal relational reasoning in the action-centric and hierarchical formulation. To this end, we introduce the Action-related Region Locator (ARL), which estimates the actionness of each spatial position to locate the action-related regions, and the Efficient Action-centric Reasoner (EAR), which can efficiently reason the temporal relations between the located regions, enabling the reasoning to be action-centric. By integrating ARL and EAR, as well as the residual structure, we propose the Action-centric Temporal-relational Reasoning (ATR) block. The overall framework is illustrated in Fig 2. Thanks to its flexibility and efficiency, we can insert the ATR block into different depths of networks with a minor computational overhead, to reason the action-centric temporal relations hierarchically. We next give details of each part.

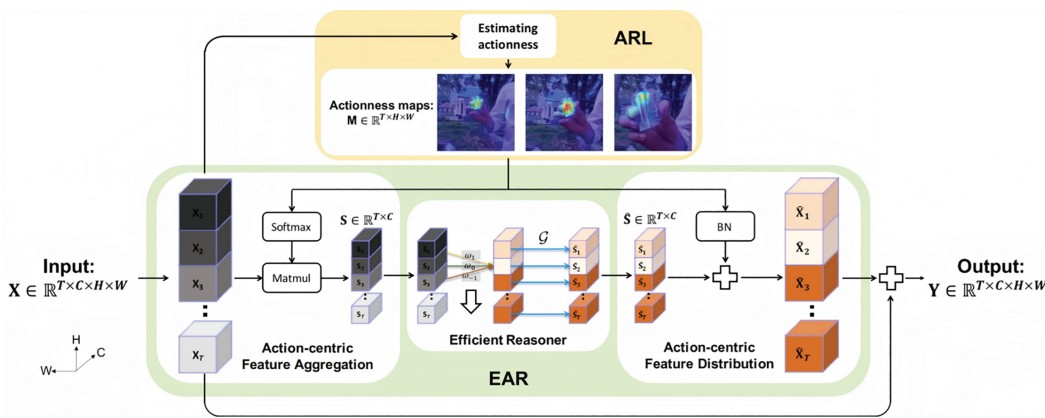

**Fig 2. The framework of the ATR block.** Given a feature map at any level, ATR first applies Action-related Region Locator (ARL) to locate action-related regions by estimating each position's actionness. The obtained actionness maps are then utilized as guidance in Efficient Action-centric Reasoner (EAR), so as to reason temporal relations action-centrically. Design details of ARL are explained in Fig 3. Note that the arrow in Efficient Reasoner means applying the same weights to **S**.

## Action-related Region Locator (ARL)

ARL aims at locating the action-related regions, so as to action-centrically reason the temporal relations later. Previous approaches leverage the object detector as the locator [24,25]. However, a major drawback is that a part of the detected objects may be unrelated to the action, causing the background distraction problem in the afterward reasoning. To tackle this problem, we propose to directly estimate the actionness, which describes the confidence of a spatial location to contain an action instance, with the ARL. The inspiration is from [50], where motion and appearance are verified as two essential visual cues to estimate actionness. Different from using auxiliary networks with RGB images and optical flow as input, ARL models motion and appearance at the feature-level to obtain actionness maps, such that action-related regions can be efficiently localized in a unified framework with the feature learning.

Fig 3 presents ARL's architecture. Concretely, given a feature map $\mathbf{X} \in \mathbb{R}^{T \times C \times H \times W}$, ARL first reduces its channel dimension from $C$ to $C_r$ by a 1×1 convolutional layer for efficiency.

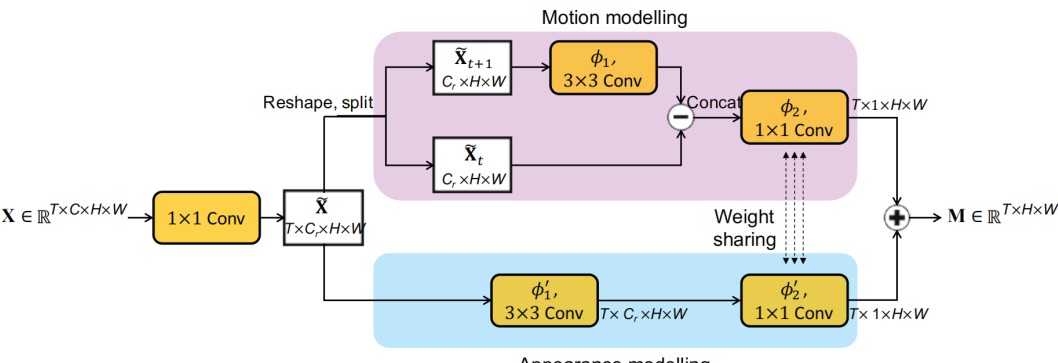

**Fig 3. The architecture of the ARL module.** Through jointly modeling motion and appearance at the feature-level, ARL can efficiently estimate the actionness of each spatial position, and realize the localization of action-related regions.

$C_r = C/r$, where $r$ is the reduction ratio and set as 16. The reduced feature map, $\tilde{\mathbf{X}}$, is next fed into:

$$\mathbf{M} = \underbrace{\mathrm{F_A}(\tilde{\mathbf{X}})}_{\text{appearance modeling}} + \underbrace{\mathrm{F_M}(\tilde{\mathbf{X}})}_{\text{motion modeling}}, \tag{1}$$

where $\mathbf{M} \in \mathbb{R}^{T \times H \times W}$ are the obtained frame-wise actionness maps. $\mathrm{F_A}$ and $\mathrm{F_M}$ separately refer to the appearance and motion modeling.

Pioneered by previous works [52,53], motion information can be extracted via transforming the temporal difference of features. Consequently, as shown in Fig 3, we construct the motion modeling as:

$$\mathrm{F_M}(\tilde{\mathbf{X}})_t = \phi_2(\phi_1(\tilde{\mathbf{X}}_{t+1}) - \tilde{\mathbf{X}}_t), \tag{2}$$

where $t$ is the temporal index. In our designs, we want the temporal difference $\phi_1(\tilde{\mathbf{X}}_{t+1}) - \tilde{\mathbf{X}}_t$ to focus on extracting the motion information, and $\phi_2$ to focus on channel contextual modeling. Accordingly, we realize $\phi_1$ as a 2D convolutional layer, with kernel size as $3 \times 3$ to better capture the motion patterns, and in depth-wise style to avoid channel interaction, making the temporal difference not distracted by other channels' contexts. The $\phi_2$ is designed as a $1 \times 1$ 2D convolutional layer, to condense the extracted motion feature and transform the channel dimension from $C/r$ to 1.

For appearance modeling, we adopt a design akin to the motion modeling. The difference is that we only consider the feature of the current time step, since the appearance can be regarded as static information. Therefore, the appearance modeling is established as: $\mathrm{F_A}(\tilde{\mathbf{X}})_t = \phi_2'(\phi_1'(\tilde{\mathbf{X}}_t))$. In order to increase model compactness, we further share the parameter between channel condensing operations in motion and appearance modeling, i.e., $\phi_2' = \phi_2$. Hence, the ultimate formulation of appearance modeling is:

$$\mathrm{F_A}(\tilde{\mathbf{X}})_t = \phi_2(\phi_1'(\tilde{\mathbf{X}}_t)). \tag{3}$$

### Efficient Action-centric Reasoner (EAR)

Upon ARL, we further propose EAR to efficiently and action-centrically reason temporal relations. EAR consists of three operations: Action-centric Feature Aggregation, Efficient Reasoner, and Action-centric Feature Distribution.

**Action-centric feature aggregation.** Guided by the actionness maps $\mathbf{M}$ gotten from ARL, we aggregate the features in action-related regions, enabling the afterward reasoning to only focus on the action itself. Specifically, given the input feature map $\mathbf{X}$ and the actionness maps $\mathbf{M}$, we first normalize $\mathbf{M}$ along spatial dimensions with *softmax* function, then frame-wisely aggregate the action-related features by calculating the weighted-sum between $\mathbf{X}$ and the normalized $\mathbf{M}$:

$$\mathbf{S}_t = \sum_{\forall i, \forall j} \frac{\exp(\mathbf{M}_t^{(i,j)})}{\sum_{\forall m, \forall n} \exp(\mathbf{M}_t^{(m,n)})} \cdot \mathbf{X}_t^{(i,j)}, \tag{4}$$

where $(i, j)$ and $(m, n)$ are both spatial indexes. $\mathbf{S}_t \in \mathbb{R}^C$ is the aggregated action-centric feature vector at time step $t$.

**Efficient reasoner.** Following [24], the temporal relations between $\mathbf{S} = \{\mathbf{S}_1, \mathbf{S}_2, ..., \mathbf{S}_T\}$ can be reasoned by the graph-based reasoner:

$$\hat{\mathbf{S}} = \mathcal{G}(\mathcal{W} \cdot \mathbf{S}), \tag{5}$$

where $\hat{\mathbf{S}}$ is the reasoned results. $\mathcal{G}$ is a channel transformation with a learnable weight of $C \times C$. $\mathcal{W} \in \mathbb{R}^{T \times T}$ is a temporal relation matrix, the element $w_{p,q}$ of which measures the correlation between time steps $p$ and $q$:

$$w_{p,q} = \frac{\exp(\langle \mathbf{W}_1 \mathbf{S}_p, \mathbf{W}_2 \mathbf{S}_q \rangle)}{\sum_q \exp(\langle \mathbf{W}_1 \mathbf{S}_p, \mathbf{W}_2 \mathbf{S}_q \rangle)}, \tag{6}$$

where $\mathbf{W}_1$ and $\mathbf{W}_2$ are learnable weights in the shape of $C \times C$.

The major limitation of Eq 6 is the huge computational overhead caused by the pair-wise interaction calculation, which makes the hierarchical reasoning infeasible. To address this issue, we propose the Efficient Reasoner, which is simplified from Eqs 6 and 6 based on the empirical observations below.

**Visualization analysis.** To intuitively understand the behavior of Eqs 6 and 6, we apply them on the ResNet-50, and evaluate it on the Something-Something V2 dataset. More details and its results are presented in Sec. Ablation study and the second line of Table 2. After training, we block-wisely average $\mathcal{W}$ cross all videos in the validation set. The results, $\overline{\mathcal{W}} = \frac{1}{N} \sum_{l=1}^{N} \mathcal{W}_l$ where $N$ is the total video number, are visualized in Fig 4(a). Intriguingly, we find that the most highlighted elements in each row are typically at the current time step and its two temporal neighbors. This pattern indicates that the reasoner tends to pay more attention to a local temporal range around the current time step.

We further investigate the distribution of each row's highlighted area: $dist(\overline{\mathcal{W}}_{t,t-1}, \overline{\mathcal{W}}_{t,t}, \overline{\mathcal{W}}_{t,t+1})$. The results are shown in Fig 4(b). Surprisingly, we observe that $dist(\overline{\mathcal{W}}_{t,t-1}, \overline{\mathcal{W}}_{t,t}, \overline{\mathcal{W}}_{t,t+1})$ at diverse time steps are extremely similar. To verify it, we calculate the Jensen–Shannon(JS) divergence between each pair of them (see S1 Appendix). The statistical results are shown at the top of Fig 4(b), where the small JS values indicate that the reasoner tends to impose similar local impact on different time steps.

**Derivation of the efficient reasoner.** Based on the above observations and analysis, we make two simplifications on $\mathcal{W}$'s calculation for efficiency: 1) Shrinkage the interaction range to be a local temporal window around the current time step. 2) Share the local distribution between different time steps. Thus, $\mathcal{W}$ is simplified as:

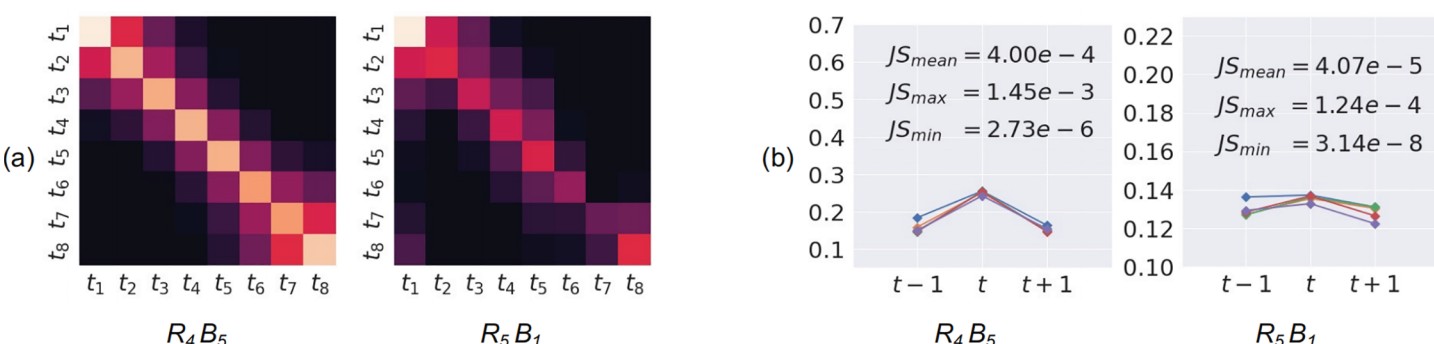

**Fig 4. (a) Visualizations of the average temporal relation matrix $\overline{\mathcal{W}}$ learned in graph-based reasoner.** (b) Distributions of $dist(\overline{\mathcal{W}}_{t,t-1}, \overline{\mathcal{W}}_{t,t}, \overline{\mathcal{W}}_{t,t+1})$ at diverse time steps $t$ in (a). JS denotes JS divergence, $S_a B_b$ denotes the $b$-th block of stage $a$. (a) and (b) indicate that the graph-based reasoner learns a similar local pattern at different time steps.

$$\begin{bmatrix} w_0 & \cdots & w_k & 0 & \cdots & 0 & 0 & \cdots & 0 \\ w_{-1} & \cdots & w_{k-1} & w_k & \cdots & 0 & 0 & \cdots & 0 \\ \vdots & \vdots & \vdots & \vdots & \ddots & \vdots & \vdots & \vdots & \vdots \\ 0 & \cdots & 0 & 0 & \cdots & w_{-k} & w_{-k+1} & \cdots & w_1 \\ 0 & \cdots & 0 & 0 & \cdots & 0 & w_{-k} & \cdots & w_0 \end{bmatrix},$$

where $k$ is a super-parameter, and $2k + 1$ is the local temporal window size.

Further, we assume the matrix $\mathcal{W}$ is instance-agnostic. Under this assumption, we can replace $\mathcal{W}$'s nonzero elements $\{w_i\}_{i=-k}^{k}$ with learnable parameters $\{\tilde{w}_i\}_{i=-k}^{k}$, deducing the final expression of the Efficient Reasoner to be:

$$\hat{\mathbf{S}}_t = \mathcal{G}\Big(\sum_{i=-k}^{k} \tilde{w}_i \cdot \mathbf{S}_{t+i}\Big), \tag{7}$$

which avoids the local interaction calculation and can be efficiently implemented by convolution.

Compared to the graph-based reasoner [24], our Efficient Reasoner significantly reduces the computation complexity and parameters. The benefit is that we can hierarchically reason the temporal relations to improve the recognition performance, by inserting the ATR block into multi-depths of a model with a slight computation overhead increase.

During implementing, we select $k$ to be 1 based on Fig 4. Following [24], we stack $n$ Efficient Reasoners to enhance the reasoning ability, with one Layer Normalization [54] followed by a ReLU as the connection of two adjacent Efficient Reasoners. $n$ is set to 2 as default.

**Action-centric feature distribution.** We finally distribute the reasoned results to each spatial location, such that the subsequent layers can heritage the enhanced temporal relational reasoning ability.

Instead of uniformly distributing via broadcasting, we action-centrically perform the distribution with the guidance of the actionness maps $\mathbf{M}$. Our intuition is that the reasoned results should be mainly distributed back to the regions where reasoning's input features are from, namely the action-related regions. Specifically, we consider two forms to instantiate the Action-centric Feature Distribution:

(a) scaling form: $\hat{\mathbf{X}}^{(i,j)} = \hat{\mathbf{S}} \times \sigma(\mathbf{M}^{(i,j)})$,
(b) adding form: $\hat{\mathbf{X}}^{(i,j)} = \hat{\mathbf{S}} + (\mathbf{BN}(\mathbf{M}))^{(i,j)}$,

where $\hat{\mathbf{X}} \in \mathbb{R}^{T \times C \times H \times W}$ is the distributed reasoned results. $\sigma$ is the *sigmoid* function. $\mathbf{BN}$ is the batch-normalization [55]. In Sec. Ablation study, we empirically find the adding form performs better. Accordingly, we choose it as default.

## ATR block and ATR-nets

The ATR block is built upon ARL and EAR. The residual structure is further introduced to ease the learning difficulty. Therefore, the output $\mathbf{Y} \in \mathbb{R}^{T \times C \times H \times W}$ of the ATR block can be expressed as: $\mathbf{Y} = \mathbf{X} + \hat{\mathbf{X}}$.

Thanks to ATR's flexible and efficient designs, we can easily insert it into standard architectures, ResNet for instance, to form ATR-nets. Specifically, we insert it right after the last **BN** of a residual block. All residual blocks in the last three stages are inserted unless stated otherwise, to explore temporal relations in rich feature hierarchies.

## Experiments

**Datasets.** We evaluate our method on three large-scale action recognition benchmarks, i.e., Something-Something V1, V2 [29], and Kinetics [28]. Something-Something V1/V2, which consists of around 108k/220k videos from 174 categories, is created to understand basic actions that occur in the physical world, and requires more attention on the temporal motion interaction of objects. In contrast, Kinetics comprises around 300k videos from 400 fine-grained and concrete categories, and scene information contributes a lot to recognizing them.

**Networks.** ResNet-50 [2] is chosen to be the baseline 2D network, since it contains no temporal relational reasoning and can isolate the effects of ATR. We also evaluate ATR on 3D network, where SlowOnly, the slow branch detached from SlowFast [31], is chosen. Compared to the standard ResNet, SlowOnly replaces the first 2D convolutional layer of all residual blocks in the last two stages with a 3D convolutional layer, the kernel size of which is set to be $3 \times 1 \times 1$. With this modification, information can be communicated over time, and the performance is significantly boosted.

**Training.** We train our models following the common protocols as stated in [5,30,31]. Specifically, we adopt distinct sampling strategies tailored to each dataset's characteristics: (1) For the Kinetics dataset where actions are more static and primarily rely on target object recognition, dense sampling is employed to capture discriminative spatial features by intensively sampling key frames; (2) For Something-Something datasets whose action categories exhibit stronger temporal dependencies, the uniform sampling strategy is utilized, which not only maintains computational efficiency but also ensures broader temporal coverage to better model long-range action evolution. These sampled $T$-frame clips are then resized to $224 \times 224$ as model inputs. The models are trained on an 8-GPU machine and each GPU has a mini-batch of 8, resulting in a total mini-batch size of 64. More training details are presented in S2 Appendix.

**Inference.** For each frame in a video clip, we follow the strategy proposed by [30,31] and resize the shorter size to 256 with maintaining the aspect ratio. Then 3 crops of $256 \times 256$ or one center crop of $224 \times 224$ that cover the full-frame are sampled for action prediction. For videos in Kinetics dataset, we uniformly sample 10 clips along their temporal axis and utilize 3-crops. For something-something datasets, we study both *accurate evaluation* (2 clips + 3 crops) and *efficient evaluation* (1 clip + 1 crop). Unless specified, we report performance under *efficient evaluation*.

### Ablation study

In this section, we conduct comprehensive ablation studies on the Something-Something V2 dataset to investigate our method. To this end, ResNet-50 is chosen as the backbone to isolate the temporal relational reasoning ability brought by ATR. The segment number $T$ is fixed to be 8. For better demonstrating, we introduce a simplified ATR block where the ARL module is removed, and name it as ATR$_{sim}$. Note that in ATR$_{sim}$, the Action-centric Feature Aggregation and Action-centric Feature Distribution of EAR module are replaced with the global average pooling and broadcasting, thus features are uniformly but not action-centrically aggregated and distributed along spatial dimensions.

**Action-centric and hierarchical formulation.** We begin by exploring the core of our method: temporal relational reasoning should be formulated in an action-centric and hierarchical manner. As shown in Table 1, the results are striking and merely inserting a single ATR block leads to a substantial performance improvement over the baseline (50.3% *v.s.* 27.5%), which highlights the effectiveness of our approach even with minimal integration.

**Table 1. Importance validation of the hierarchical, action-centric formulation for temporal relational reasoning: S{i} means inserting in all blocks of ResNet's i-th stage [2], and S5-3 means inserting in the 3-rd block of the 5-th stage.** $\text{ATR}_{sim}$ **denotes a simplified ATR block without action-centric formulation.**

|  |  | Extra layers | Top-1 | Top-5 |
|---|---|---|---|---|
| TSN-Res50 (our impl.) |  | 0 | 27.5 | 56.2 |
| +$\text{ATR}_{sim}$ |  | S5-3 | 47.1 | 75.0 |
| +ATR | S5-3 | 1 | 50.3 | 77.7 |
| +ATR | S{5} | 3 | 56.5 | 82.7 |
| +ATR | S{4,5} | 9 | 57.8 | 83.8 |
| +ATR | S{3,4,5} | 13 | **59.6** | 85.6 |
| +ATR | S{2,3,4,5} | 16 | 59.5 | **85.8** |

Furthermore, the significant accuracy gap between ATR and $\text{ATR}_{sim}$ (50.3% *v.s.* 47.1%) demonstrates the crucial role of the action-centric formulation in enhancing temporal reasoning.

Building on this foundation, we progressively insert additional ATR blocks into the model. As expected, the performance continues to improve, illustrating the advantage of adopting a hierarchical structure for temporal relational reasoning. This progression supports our hypothesis that deeper integration of ATR blocks enables the model to better capture long-range dependencies and fine-grained temporal relationships.

However, we observe that performance improvements begin to plateau once we insert ATR blocks into the second stage. This saturation effect suggests that, after a certain point, the model may have already sufficiently captured the necessary temporal relationships and further additions offer diminishing returns. In light of this, we have chosen to insert ATR blocks only into the last three stages as a default configuration, balancing both accuracy and computational efficiency. This trade-off ensures that the model achieves optimal performance without incurring unnecessary computational overhead.

**Effectiveness of the efficient reasoner.** We compare our proposed Efficient Reasoner with the graph-based reasoner [24] based on $\text{ATR}_{sim}$ and under the case of no stacking. The results are given in Table 2. Distinctly, our Efficient Reasoner outperforms the graph-based reasoner by 5.5% in accuracy while introducing 90% less computational overhead, demonstrating its clear superiority in both performance and efficiency. This highlights the advantage of our design, which avoids the need for expensive graph construction and message passing, and instead leverages lightweight attention mechanisms to capture temporal dependencies in a more scalable and flexible manner.

We further study the effect of stacking multiple Efficient Reasoners. As shown in Table 2, the performance can be improved when stacking two Efficient Reasoners (54.7% *v.s.* 53.2%), but the improvement saturates when stacking more. Consequently, we stack two Efficient Reasoners in the EAR module as default.

**Table 2. Efficient reasoner (ER) *v.s.* graph-based reasoner (GR): Our ER outperforms GR from both accuracy and efficiency perspectives. Note that GR/ER×*n* means stacking *n* GRs/ERs.**

|  | Reasoner | Extra GFLOPs | Top-1 | Top-5 |
|---|---|---|---|---|
| TSN-Res50 | - | 0 | 27.5 | 56.2 |
| +$\text{ATR}_{sim}$ | GR×1 | 0.50 | 47.7 | 76.8 |
| +$\text{ATR}_{sim}$ | ER×1 | 0.05 | 53.2 | 80.2 |
| +$\text{ATR}_{sim}$ | ER×2 | 0.08 | 54.7 | **81.4** |
| +$\text{ATR}_{sim}$ | ER×3 | 0.10 | **54.8** | 81.1 |

**Table 3. Motion and appearance modeling in ARL: Both are crucial. App.: appearance, Mo.: motion. AFA is short for the Action-centric Feature Aggregation.**

| TSN-Res50 + ATR$_{sim}$ | ARL | Top-1 | Top-5 |
|---|---|---|---|
| - | - | 54.7 | 81.4 |
| +AFA | App. | 56.0 | 82.8 |
| +AFA | Mo. | 57.0 | 83.5 |
| +AFA | Mo.+App. | **57.7** | **84.2** |

**Design details of ARL.** To explore the design details of the ARL module, we activate the Action-centric Feature Aggregation operation in the EAR module over ATR$_{sim}$. As shown in Table 3, both motion and appearance cues are essential for estimating actionness, which aligns with the findings in [50]. Specifically, appearance features help in identifying the visual context of the scene, such as objects and actors, while motion features capture temporal changes crucial for understanding action dynamics. Their complementary nature ensures more accurate localization of action-relevant regions, thereby providing a reliable foundation for downstream reasoning. Furthermore, the notable performance gain after enabling the Action-centric Feature Aggregation operation once again validates the importance of our action-centric formulation. This improvement suggests that explicitly modeling the spatial regions relevant to action, rather than treating the entire frame uniformly, enables the model to concentrate its reasoning capacity where it matters most. It helps suppress irrelevant background noise and emphasizes semantically meaningful regions over time, which is particularly beneficial in datasets where fine-grained motion understanding is crucial. These findings reinforce our design choice of integrating both appearance and motion cues into the ARL module and confirm the effectiveness of action-centric feature aggregation as a core mechanism to drive accurate and efficient temporal relational reasoning.

**Action-centric feature distribution's two forms.** We further activate the Action-centric Feature Distribution operation to investigate its design details. The results are listed in Table 4. As expected, action-centrically distributing the reasoned feature is superior to uniformly distributing. Besides, the adding form performs better than the scaling form when instantiating the Action-centric Feature Distribution (59.6% *v.s.* 58.9%). We suppose the reason is that adding form can alleviate the learning difficulty of ARL and EAR modules, since their deviates are independent with each other during back-propagation. In contrast, when scaling form is adopted, deviates of two modules are entangled because of multiplication, burdening the learning of both.

## Generality exploration

Benefiting from the lightweight and flexible designs, ATR can be seamlessly inserted into a normal CNN to enhance its ability of reasoning temporal relations. In this section, we investigate various CNN baselines, and demonstrate our proposed ATR can significantly improve their performances on multiple action recognition datasets.

**Table 4. Two forms of the Action-centric Feature Distribution (AFD): Adding form performs better.**

| TSN-Res50 + ATR$_{sim}$ | AFD's form | Top-1 | Top-5 |
|---|---|---|---|
| +AFA | - | 57.7 | 84.2 |
| +AFA+AFD | Scaling | 58.9 | 84.8 |
| +AFA+AFD | Adding | **59.6** | **85.6** |

Table 5 provides the results of $3 \times 3$ experiments: Three baselines which are TSN-Res50, TSM-Res50, and SlowOnly-Res50, and three datasets including Kinetics, Something-Something V1, and V2. The frame number $T$ is herein fixed to be 8. As expected, the performances under all settings are consistently and remarkably boosted after equipping ATR, demonstrating its effectiveness and generality. Moreover, though TSN-Res50 provides relatively poor performance, after equipping, it becomes competitive or even superior to TSM-Res50 and SlowOnly-Res50. Integrating ATR in these two SOTA networks further brings notable gains. These results fully demonstrate the power of our hierarchical and action-centric temporal relational reasoning, which is hardly covered in previous works.

## Comparisons with the state-of-the-arts

In this section, we compare ATR to other state-of-the-art temporal relational reasoning methods and show our ATR surpasses them in both accuracy and efficiency aspects. We also give comparisons with other published approaches (see S3 Appendix), where ATR still achieves SOTA performances, confirming its superiority.

**Something-something V1 & V2.** The results are given in Table 6, where TSM+ATR's performances are reported under *accurate evaluation*. Explicitly, our ATR is superior to all the other temporal relational reasoning methods: 1) Compared to the ones without using the detector, i.e. TRN [17] and TRG [56], our ATR outperforms them respectively by 16.1%/15.2% and 5.8%/5.6%, on Something-Something V1/V2 under the same frame setting. 2) Compared to the ones relying on the detector, our ATR not only performs better but also is significantly more efficient, since ATR only involves lightweight operators but not the heavy detector. These results demonstrate ATR's superiority, along with the advantages of the action-centric and hierarchical formulation for temporal relational reasoning.

**Kinetics.** Table 7 presents the overall results. It is worth mentioning that out of all the other temporal relational reasoning methods, only MUSLE [27] provides the results on Kinetics. A major reason we suppose is that Kinetics contains numerous human-centric videos with

**Table 5. Improvements over diverse baselines on Kinetics, Something-Something V1 & V2 datasets. Evaluation protocols: 3 crops + 10 clips/video for Kinetics, 1 crop + 1 clips/video for Something-Something (Sth).**

| Dataset | Model | Top-1 | Top-5 | ΔTop-1 |
|---|---|---|---|---|
| Kinetics | TSN-Res50 | 70.5 | 89.2 | **+4.1** |
| Kinetics | +ATR | 74.6 | 91.5 | **+4.1** |
| Kinetics | TSM-Res50 | 74.1 | 91.2 | **+1.6** |
| Kinetics | +ATR | 75.7 | 92.0 | **+1.6** |
| Kinetics | SlowOnly-Res50 | 74.9 | 91.9 | **+1.4** |
| Kinetics | +ATR | 76.3 | 92.6 | **+1.4** |
| Sth-V1 | TSN-Res50 | 19.5 | - | **+24.2** |
| Sth-V1 | +ATR | 43.7 | 72.1 | **+24.2** |
| Sth-V1 | TSM-Res50 | 45.6 | 74.2 | **+3.4** |
| Sth-V1 | +ATR | 49.0 | 77.9 | **+3.4** |
| Sth-V1 | SlowOnly-Res50 | 46.6 | 75.1 | **+1.0** |
| Sth-V1 | +ATR | 47.6 | 75.9 | **+1.0** |
| Sth-V2 | TSN-Res50 | 27.5 | 56.2 | **+32.1** |
| Sth-V2 | +ATR | 59.6 | 85.6 | **+32.1** |
| Sth-V2 | TSM-Res50 | 59.1 | 85.6 | **+2.6** |
| Sth-V2 | +ATR | 61.7 | 86.7 | **+2.6** |
| Sth-V2 | SlowOnly-Res50 | 58.5 | 84.6 | **+2.4** |
| Sth-V2 | +ATR | 60.9 | 86.5 | **+2.4** |

**Table 6. Comparisons with the state-of-the-arts on Something-Something (Sth) V1 and V2. "N/A" indicates the numbers are not available. "*" means the object detector is leveraged.**

| Method | Backbone | Frames | Params | GFLOPs×Clips×Crops | Top-1@Sth-V1 | Top-1@Sth-V2 |
|---|---|---|---|---|---|---|
| TRN$_{multiscale}$ [17] | BNInception | 8 | - | N/A×1×1 | 34.4 | 48.8 |
| TRG [56] | ResNet-50 | 16 | 28.4M | 66×2×3 | 48.1 | 59.8 |
| TRG [56] | ResNet-50 | 32 | 28.4M | 132×2×3 | 49.5 | 62.2 |
| NL I3D+GCN* [24] | 3D ResNet-50 | 32 | 62.2M | 303×2×1 | 46.1 | - |
| STIN* [25] | 3D ResNet-50 | 16 | - | N/A×N/A×N/A | - | 60.2 |
| ASS* [26] | ResNet-50 | 32 | - | N/A×N/A×N/A | 51.4 | 63.5 |
| MUSLE* [27] | ResNet-50 | 16 | - | N/A×N/A×N/A | 52.5 | 65.0 |
| TimeFormer [57] | Transformer | 16 | 121.4M | N/A×N/A×N/A | - | 62.2 |
| TSM+ATR (Ours) | ResNet-50 | 8 | 25.5M | 34×2×3 | **50.5** | **64.0** |
| TSM+ATR (Ours) | ResNet-50 | 16 | 25.5M | 69×2×3 | **53.9** | **65.4** |

**Table 7. Comparisons with the state-of-the-arts on Kinetics. One view refers to one temporal clip with one spatial crop.**

| Methods (2D-CNNs) | Backbone | Frames | GFLOPs × Views | Top-1 |
|---|---|---|---|---|
| MUSLE [27] | ResNet-50 | 8 | N/A×N/A | 75.1 |
| TSM+ATR (Ours) | ResNet-50 | 8 | 34×30 | **75.7** |

complex backgrounds, and some action categories involve only a few relevant objects. As a result, scene understanding becomes more critical, while reasoning about object transformations over time is less beneficial. Nevertheless, the proposed action-centric feature aggregation in ARL can effectively localize action-relevant regions and enhance the network's ability to understand scenes. Therefore, our method achieves better performance in temporal relational reasoning compared to MUSLE.

## Visualization

To verify that our ATR can concentrate on action-related regions hence realize the overall action-centric formulation as we claim, we further visualize and analyze the learned action-ness maps, **M**, from the TSM-Res50+ATR model.

The visualization results of several examples are shown in Fig 5. Distinctly, we can see that our ATR provides decent results: Action-related regions are highlighted as expected, while background regions are almost with no response. Besides, our ATR can automatically emphasize the frame when the action is ongoing, while reason nearly nothing when it is out of the action's temporal range. Take the second example, *Moving something away from something*, for concrete demonstration. Though the bottle, towel, the column moved by hand, and tissue paper are closed to each other, our ATR only presents high responses on the column and tissue paper, which are action-related, while nearly zero-response on the bottle and towel, which are unrelated to this action. Moreover, high responses are shown in the first two frames, when the column is being moved by hand. On the contrary, in the last two frames when the moving is finished, our ATR's responses turn to be significantly weak. These results and observations strongly support our action-centric motivation and designs.

## Limitation

Although our proposed approach achieved excellent performance gains for various backbones on three action recognition benchmarks, we also considered the potential limitations, including: 1) Transferability to highly complex scenes or multi-action scenarios. 2) The

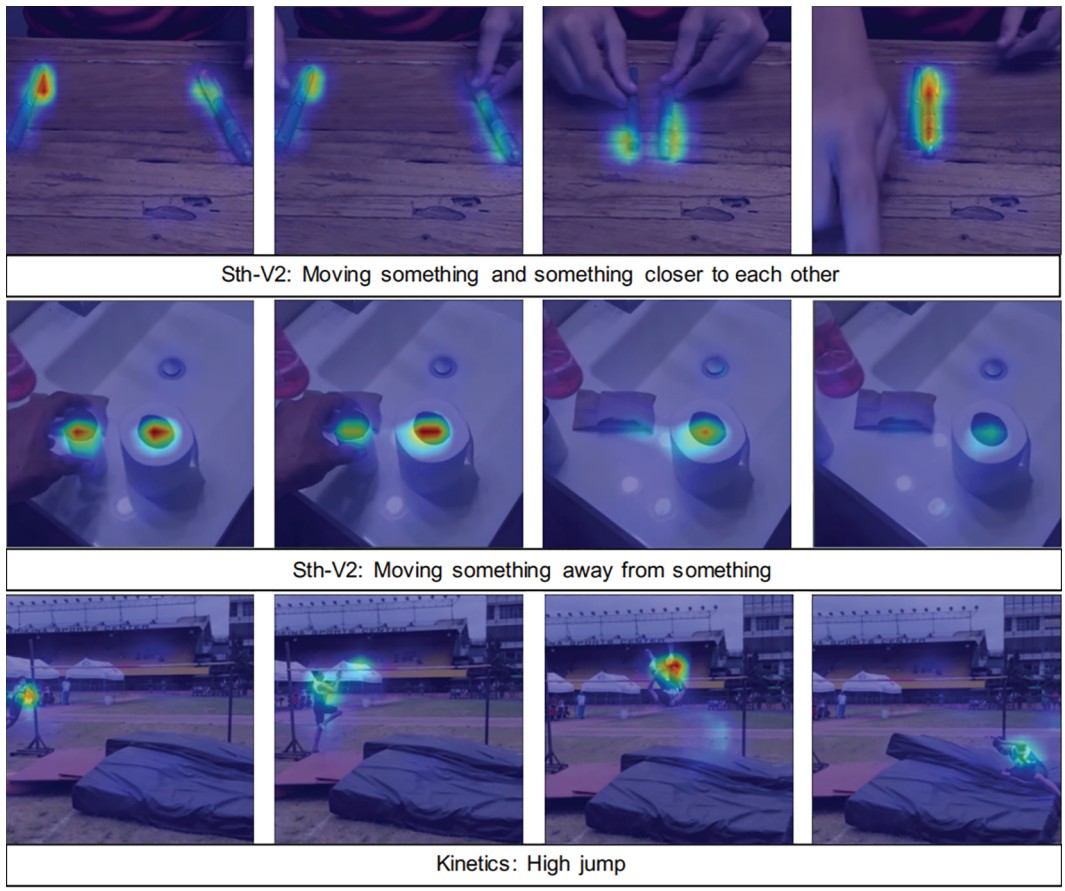

**Fig 5. Visualizations of the learned actionness maps, M.** Obviously, our ATR can sufficiently focus on the action-related regions.

efficient reasoner lacks spatial information. To avoid the significant computational overhead caused by pairwise interaction calculations in the spatial dimension, the efficient reasoner performs temporal relational reasoning along the channel dimension, which may lead to reduced performance.

## Conclusions

In this work, we propose an action-centric and hierarchical formulation to reason the temporal relations for action recognition. We introduce a generic Action-centric Temporal-relational Reasoning (ATR) block, which comprises an Action-related Region Locator (ARL) and an Efficient Action-centric Reasoner (EAR). Building upon the ATR block, we further develop ATR-nets, which can be seamlessly integrated into various backbones to enable hierarchical and efficient temporal relational reasoning. Extensive experiments across multiple benchmarks (Something-Something V1/V2 and Kinetics) and backbones (TSN, TSM, SlowOnly) demonstrate the effectiveness, generality, and efficiency of our approach. In particular, our method consistently outperforms strong baselines with minor computational overhead. Moreover, visualizations and ablation studies further support our design motivation and validate the contribution of each module.

## Supporting information

**S1 Appendix. Jensen–Shannon divergence.**
(PDF)

**S2 Appendix. More training details.**
(PDF)

**S3 Appendix. More state-of-the-art comparisons.**
(PDF)

## Author contributions

**Conceptualization:** Manshu Liang.

**Data curation:** Manshu Liang.

**Methodology:** Manshu Liang, Zhuolei Chen.

**Validation:** Wenbin Wu, Tengfei Han.

**Visualization:** Manshu Liang, Tengfei Han.

**Writing – original draft:** Manshu Liang, Tengfei Han.

**Writing – review & editing:** Zhuolei Chen, Yuan Zheng.

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
