## [Decision Letter · Decision Letter 0]

2 May 2025

PONE-D-25-20267Reasoning Action-centric Temporal Relations at Rich Feature Hierarchies for Action RecognitionPLOS ONE

Dear Dr. Liang,

Thank you for submitting your manuscript to PLOS ONE. After careful consideration, we feel that it has merit but does not fully meet PLOS ONE’s publication criteria as it currently stands. Therefore, we invite you to submit a revised version of the manuscript that addresses the points raised during the review process.

We look forward to receiving your revised manuscript.

Kind regards,

Hikmat Ullah Khan, PhD (Computer Science)

Academic Editor

PLOS ONE

5. We are unable to open your Supporting Information file [egbib, egbib2, plos and splncs04]. Please kindly revise as necessary and re-upload.

Additional Editor Comments:

Dear Authors, overall good work, as per comments by three reviewers. However, let me share few comments

Recently publications of top journals are expected to be discussed in related work.

There is not a single paper of year 2025. How let me suggest that the proposed work should be compared with following paper:

ResLNet: deep residual LSTM network with longer input for action recognition. Frontiers of Computer Science, 16(6), 166334. doi: https://doi.org/10.1007/s11704-021-0236-9

In addition, few more should be discussed in related work, such as

Nie, F., et al (2025). An Adaptive Solid-State Synapse with Bi-Directional Relaxation for Multimodal Recognition and Spatio-Temporal Learning. Advanced Materials, 2412006. doi: https://doi.org/10.1002/adma.202412006

TASTA: Text-Assisted Spatial and Temporal Attention Network for Video Question Answering. Advanced Intelligent Systems, 5(4), 2200131. doi: https://doi.org/10.1002/aisy.202200131

Reviewers' comments:

Reviewer's Responses to Questions

**Comments to the Author**

1. Is the manuscript technically sound, and do the data support the conclusions?

Reviewer #1: Partly

Reviewer #2: Yes

Reviewer #3: Yes

2. Has the statistical analysis been performed appropriately and rigorously? 

Reviewer #1: No

Reviewer #2: Yes

Reviewer #3: Yes

3. Have the authors made all data underlying the findings in their manuscript fully available?

Reviewer #1: Yes

Reviewer #2: Yes

Reviewer #3: No

4. Is the manuscript presented in an intelligible fashion and written in standard English?

Reviewer #1: Yes

Reviewer #2: Yes

Reviewer #3: Yes

5. Review Comments to the Author

Reviewer #1: The paper is of excellent quality, presenting valuable insights through a thorough analytical approach. It demonstrates a well-structured methodology and yields relevant and meaningful findings within the study's defined scope. The proposed Action-centric Temporal-relational Reasoning (ATR) block is a compelling contribution, with clear practical implications in action recognition.

1. Title and Abstract: The title effectively captures the study's focus on reasoning action-centric temporal relations in hierarchical feature spaces, ensuring clarity and relevance. However, a slight refinement could further highlight the practical or application-level implications of the work. The abstract provides a concise overview of the research objectives, methodology, and key findings, but incorporating a brief mention of the practical significance or deployment scenarios of the results would enhance its impact.

2. Introduction and Related work: The introduction is well-structured and clearly outlines the research problem while contextualizing the literature within existing studies. However, to strengthen its relevance and depth, the authors should articulate the research objectives more precisely and clearly highlight the specific research gap.

3. The methodology section is well-structured, with clear explanations of the ARL and EAR components and their integration into the ATR block. In the experiments section, the distinction between the temporal focus of Something-Something and the scene-driven nature of Kinetics is clear; however, the authors should more explicitly justify the choice of these datasets in relation to their method’s objectives. The explanation of SlowOnly’s 3D convolution modification is clear, though a brief rationale for the choice of the 3×1×1 kernel would enhance understanding of its impact on temporal modeling. The use of dense sampling for Kinetics and uniform sampling for Something-Something is appropriate, but adding a one-line justification for each (e.g., balancing efficiency vs. temporal coverage) would improve clarity. Additionally, the phrase “3-crops of 256 × 256” could be clarified does cropping occur after resizing the short side to 256?

4. The ablation study section is systematically presented; however, the authors should revise and expand the interpretations to provide more detailed and insightful analysis.

5. Author should strongly revise and extend the conclusions section.

Reviewer #2: Author Comment

The authors propose a novel Action-centric Temporal-relational Reasoning (ATR) block for enhancing action recognition by focusing on action-centric and hierarchical temporal relation reasoning. The work is well-organized, the methodology is clear, and the experimental results convincingly demonstrate the proposed method's effectiveness across multiple datasets and baseline architectures. Visualizations further support the claims. Overall, this work addresses a critical limitation in current methods (background distraction and lack of multi-level reasoning) and offers a promising solution.

Strengths:

• Innovative formulation of action-centric and hierarchical temporal reasoning.

• Strong experimental validation across multiple datasets (Something-Something V1/V2, Kinetics).

• Clear methodology with detailed ablation studies and insightful visualizations.

• Lightweight and flexible design, easily pluggable into existing architectures.

Weaknesses and Suggestions:

• Comparison with newer models: It would strengthen the paper to benchmark against recent transformer-based methods (e.g., TimesFormer, VideoMAE).

• Efficiency Reporting: In addition to FLOPs, providing inference time and parameter size would give a fuller view of the model’s efficiency.

• Limitations Discussion: A brief mention of potential limitations (e.g., transferability to highly complex scenes or multi-action scenarios) would add balance.

• Kinetics Analysis: Since Kinetics is less motion-focused, a more detailed discussion on ATR's improvements there would be valuable.

Minor Points (Typos/Formatting):

• Line 5: "academic and industrial communities." Consider adding a reference supporting this claim.

• Fig. 2 caption: "Design details of ARL are demonstrated in Fig. 3" maybe say "explained" instead of "demonstrated" for better English flow.

• Page formatting: Be cautious of occasional broken lines across pages (e.g., equations (1), (2), etc.).

• References: The style is slightly inconsistent in some parts (e.g., mixing arXiv preprints with conference papers); better formatting may be needed for final publication.

N. B.: Very good work; addressing the above points will further enhance the paper's quality.

Reviewer #3: This paper presents a new method called ATR for video action recognition, which focuses on identifying action-related regions in a video and reasoning about how they change over time. The ATR block includes two main parts: the Action-related Region Locator (ARL) and the Efficient Action-centric Reasoner (EAR). The method is lightweight, can be easily added to different models, and is tested on three major datasets (Something-Something V1, V2, and Kinetics) using popular backbones like TSN, TSM, and SlowOnly. The experiments are well designed, with detailed ablation studies and visualizations that support the method’s design and effectiveness. The results show consistent performance improvements, demonstrating both the accuracy and general usefulness of ATR.

However, there are some areas that need improvement. The writing needs better grammar and clarity in several places, with some phrases being unclear. The paper does not compare its method to transformer-based video models, which are important recent baselines. Some key settings (like window size and number of layers) are fixed without testing how much they affect the results. Also, the paper does not mention any possible limitations, such as how the method performs on long or complex videos.

Finally, while the datasets are public, there is no full Data Availability Statement to confirm whether code or outputs will be shared.

With some minor revisions and clearer discussion, this paper will make a strong contribution to the field.

6. PLOS authors have the option to publish the peer review history of their article (what does this mean?). If published, this will include your full peer review and any attached files.

Reviewer #1: No

Reviewer #2: No

Reviewer #3: No

---

## [Author Response · Author response to Decision Letter 1]

20 May 2025

Response to Associate Editor

Comment: Overall good work, as per comments by three reviewers. However, let me share a few comments. Recently publications of top journals are expected to be discussed in related work. There is not a single paper of year 2025. How let me suggest that the proposed work should be compared with the following paper: ResLNet: deep residual LSTM network with longer input for action recognition. In addition, a few more should be discussed in related work, such as Nie, F., et al (2025). An Adaptive Solid-State Synapse with Bi-Directional Relaxation for Multimodal Recognition and Spatio-Temporal Learning. TASTA: Text-Assisted Spatial and Temporal Attention Network for Video Question Answering.

Reply: Thank you for your constructive comments on our manuscript. We agree that incorporating recent publications is important for improving the completeness and relevance of the related work. In the revised manuscript, we have added discussions of several recent studies. These works are now included in the Related Work section (see lines 73–79, 88–90, 103–104, and 119–121). We also fully addressed all review- ers’ concerns in the revised version. We sincerely hope that this revised manuscript will be considered for publication.

Response to Referee #1

Comment 1: The abstract provides a concise overview of the research objectives, methodology, and key findings, but incorporating a brief mention of the practical significance or deployment scenarios of the results would enhance its impact.

Reply: Thank you for your insightful comment. In the revised manuscript, we have revised the abstract to include a brief statement on the practical relevance of our proposed method.

Comment 2: Introduction and Related work: The introduction is well-structured and clearly outlines the research problem while contextualizing the literature within existing studies. However, to strengthen its relevance and depth, the authors should articulate the research objectives more precisely and clearly highlight the specific research gap.

Reply: Thank you for your insightful suggestion. In response, we have revised the introduction to more clearly articulate the research objectives and highlight the specific research gap. In particular, we emphasize that existing works relying on object detectors still suffer from background interference due to their inclusion of irrelevant objects. Additionally, we point out that these methods typically overlook low-level features and are difficult to integrate into multi-level backbones due to their heavy computational overhead and

structural inflexibility. To address these issues, we now explicitly state our goal: to formulate temporal relational reasoning in an action-centric and hierarchical manner—focusing on (1) suppressing background distractions by accurately locating action-relevant regions, and (2) enabling efficient integration with multi- level feature hierarchies. The revised paragraph can be found in the Introduction section (lines 21–33, page 2) of the updated manuscript.

Comment 3: The use of dense sampling for Kinetics and uniform sampling for Something-Something is appropriate, but adding a one-line justification for each (e.g., balancing efficiency vs. temporal coverage) would improve clarity. Additionally, the phrase “3-crops of 256 × 256” could be clarified does cropping occur after resizing the short side to 256?

Reply: Thank you for your suggestion. We have addressed your comment by adding one-line justifications for the sampling strategies used for Kinetics and Something-Something datasets. Specifically, we clarify that dense sampling is adopted for Kinetics due to its relatively static actions, where capturing discriminative spatial details is crucial. In contrast, uniform sampling is used for Something-Something, which contains temporally structured actions, to ensure broader temporal coverage while maintaining computational effi- ciency (lines 269-276, page 8). In addition, we have revised the phrase “3-crops of 256 × 256” for clarity. We now explicitly state that the cropping is performed after resizing the shorter side of the frame to 256 pixels (lines 280-283, page 8).

Comment 4: The ablation study section is systematically presented; however, the authors should revise and expand the interpretations to provide more detailed and insightful analysis.

Reply: Thank you for your suggestion. Following your suggestion, we have carefully revised and expanded our interpretations to provide more detailed and insightful analysis. The revised explanations can be found in lines 298–317, lines 321-326, and lines 332-349.

Comment 5: The author should strongly revise and extend the conclusions section.

Reply: Thank you for your suggestion. In response, we have thoroughly revised and extended the conclusion to better highlight the contributions and findings of our work. Please see the lines 426-437 on page 13.

Response to Referee #2

Comment 1: Comparison with newer models: It would strengthen the paper to benchmark against recent transformer-based methods (e.g., TimesFormer, VideoMAE)

Reply: Thank you for the suggestion. We have added the performance of TimesFormer in Table 6 (page 12). For VideoMAE, we do not include a direct comparison, because it is a video masked autoencoder that employs a self-supervised learning approach to obtain more effective video representations during pre-training. In contrast, our proposed method adopts a supervised learning strategy.

Comment 2: Efficiency Reporting: In addition to FLOPs, providing inference time and parameter size would give a fuller view of the model’s efficiency.

Reply: Thank you for the valuable suggestion. In addition to reporting FLOPs, we have now included the parameter size of our model in the revised manuscript (see Table 6 on page 12).

Comment 3: Limitations Discussion: A brief mention of potential limitations (e.g., transferability to highly complex scenes or multi-action scenarios) would add balance.

Reply: Thank you for the suggestion. We have added a discussion of the limitations in the revised version (lines 380–387, page 12).

Comment 4: Kinetics Analysis: Since Kinetics is less motion-focused, a more detailed discussion on ATR’s improvements there would be valuable.

Reply: Thank you for the suggestion. As suggested by the reviewer, we have added more discussions on ATR’s improvements. (lines 359–366, page 11)

Comment 5: Line 5: “academic and industrial communities.” Consider adding a reference supporting this claim.

Reply: We sincerely appreciate the valuable comments. We have added two references after “academic and industrial communities”.

1.Wang M, Xing J, Jiang B, et al. A multimodal, multi-task adapting framework for video action recogni- tion[C]. AAAI. 2024, 38(6): 5517-5525.

2.Chen J, Ho C M. MM-ViT: Multi-modal video transformer for compressed video action recognition[C]. WCACV. 2022: 1910-1921.

Comment 6: Fig. 2 caption: “Design details of ARL are demonstrated in Fig. 3” maybe say “explained” instead of “demonstrated” for better English flow.

Reply: Thank you for the helpful suggestion. We have revised the caption of Fig. 2 by replacing “demon- strated” with “explained” to improve the clarity and fluency of the expression.

Comment 7: Page formatting: Be cautious of occasional broken lines across pages (e.g., equations (1), (2), etc.).

Reply: Thank you for the suggestion. We have addressed this in the revised manuscript.

Comment 8: References: The style is slightly inconsistent in some parts (e.g., mixing arXiv preprints with conference papers); better formatting may be needed for final publication.

Reply: Thank you for the suggestion. We have checked the references carefully and corrected the errors accordingly in the revised manuscript.

Response to Referee #3

Comment 1: The writing needs better grammar and clarity in several places, with some phrases being unclear.

Reply: Thank you for your suggestion. We have carefully polished the language in the revised manuscript to enhance clarity and readability.

Comment 2: The paper does not compare its method to transformer-based video models, which are impor- tant recent baselines.

Reply: Thank you for the suggestion. We have added the performance of TimesFormer in Table 6 (page 12).

Comment 3: Some key settings (like window size and number of layers) are fixed without testing how much they affect the results.

Reply: Thank you for the suggestion. Table 1 presents the results with different numbers of ATR blocks. We observe a consistent performance improvement as more ATR blocks are inserted. For the window size k, as discussed in the Visualization Analysis (line 197, page 6), the most activated elements in each row are typically centered around the current time step and its two adjacent temporal neighbors. Additionally, the low Jensen–Shannon divergence values suggest that the reasoner applies a relatively consistent local influence across time steps. Thus, increasing the window size k does not lead to further performance gains.

Comment 4: The paper does not mention any possible limitations, such as how the method performs on long or complex videos.

Reply: Thank you for your valuable comment regarding the discussion of potential limitations. In the revised manuscript, we have added a specific paragraph (lines 417–424, page 13) to explicitly address the limitations of our proposed method. While our approach achieves strong performance across multiple backbones and datasets, we acknowledge two main limitations: 1) Transferability to complex or multi-action scenarios. 2) Lack of spatial relational reasoning in the Efficient Reasoner.

---

## [Decision Letter · Decision Letter 1]

13 Jun 2025

Reasoning Action-centric Temporal Relations at Rich Feature Hierarchies for Action Recognition

PONE-D-25-20267R1

Dear Dr. Liang,

We’re pleased to inform you that your manuscript has been judged scientifically suitable for publication and will be formally accepted for publication once it meets all outstanding technical requirements.

Kind regards,

Hikmat Ullah Khan, PhD (Computer Science)

Academic Editor

PLOS ONE

Additional Editor Comments (optional):

Reviewers' comments:

Reviewer's Responses to Questions

**Comments to the Author**

1. If the authors have adequately addressed your comments raised in a previous round of review and you feel that this manuscript is now acceptable for publication, you may indicate that here to bypass the “Comments to the Author” section, enter your conflict of interest statement in the “Confidential to Editor” section, and submit your "Accept" recommendation.

Reviewer #2: All comments have been addressed

Reviewer #4: All comments have been addressed

2. Is the manuscript technically sound, and do the data support the conclusions?

Reviewer #2: Yes

Reviewer #4: Yes

3. Has the statistical analysis been performed appropriately and rigorously? 

Reviewer #2: Yes

Reviewer #4: Yes

4. Have the authors made all data underlying the findings in their manuscript fully available?

Reviewer #2: Yes

Reviewer #4: Yes

5. Is the manuscript presented in an intelligible fashion and written in standard English?

Reviewer #2: Yes

Reviewer #4: Yes

6. Review Comments to the Author

Reviewer #2: Reviewer Recommendation: Accept

The authors have thoughtfully and clearly answered every one of my recommendations. Significant improvements have been made to the book, including the addition of current standards (TimesFormer), enlarged efficiency measures, and a fair assessment of its limitations. Language, formatting, and reference updates, together with clarifications regarding ATR's Kinetics performance, further increase its quality.

The updated manuscript exhibits clarity and technical rigor. Acceptance is what I advise

Reviewer #4: 1- New experiments and comparisons added. TimesFormer has been inserted as a transformer-based baseline; model‐size statistics now accompany FLOPs; ablation comments were expanded.

2- Methodological clarifications provided. Sampling strategies, window-size rationale and efficiency trade-offs are now explicitly justified.

3- Limitations acknowledged. Section on transferability to complex scenes and lack of spatial reasoning added.

Overall, I see that the authors have addressed the reviewers comments on their manuscript.

7. PLOS authors have the option to publish the peer review history of their article (what does this mean?). If published, this will include your full peer review and any attached files.

Reviewer #2: No

Reviewer #4: No

---

## [Editor Report · Acceptance letter]

PONE-D-25-20267R1

PLOS ONE

Dear Dr. Liang,

I'm pleased to inform you that your manuscript has been deemed suitable for publication in PLOS ONE. Congratulations! Your manuscript is now being handed over to our production team.

Kind regards,

on behalf of

Dr. Hikmat Ullah Khan

Academic Editor

PLOS ONE